# Tracking the Genomic Evolution of SARS-CoV-2 for 29 Months in South Korea

**DOI:** 10.3390/v15040873

**Published:** 2023-03-29

**Authors:** Seri Jeong, Jae-Seok Kim, Su Kyung Lee, Eun-Jung Cho, Jungwon Hyun, Wonkeun Song, Hyun Soo Kim

**Affiliations:** 1Department of Laboratory Medicine, Hallym University Kangnam Sacred Heart Hospital, Hallym University College of Medicine, Anyang 14068, Republic of Korea; 2Department of Laboratory Medicine, Kangdong Sacred Heart Hospital, Hallym University College of Medicine, Anyang 14068, Republic of Korea; 3Department of Laboratory Medicine, Hallym University Dongtan Sacred Heart Hospital, Hallym University College of Medicine, Anyang 14068, Republic of Korea

**Keywords:** SARS-CoV-2, COVID-19, genome, sequencing, evolution, variant, phylogeny, Korea

## Abstract

The pandemic caused by the severe acute respiratory syndrome coronavirus 2 (SARS-CoV-2) has continued, with the persistent emergence of variants of concern (VOCs). Therefore, this study aimed to track the genomic evolution of SARS-CoV-2 strains by sequencing the spike protein for 29 months, which accounted for the majority of the COVID-19 pandemic period. A total of 109 swabs from patients with confirmed coronavirus disease 2019 (COVID-19) infection were randomly collected between March 2020 and July 2022. After genomic sequencing, we analyzed the naming systems and phylogenetic trees. Five surge peaks of COVID-19 cases have been reported in South Korea, resulting in 14,000,000 cumulative confirmed cases and 17,000 deaths. Among the sequenced samples, 34 wild-type strains and 75 VOCs, including 4 Alpha, 33 Delta, 2 Epsilon, and 36 Omicron VOCs, were identified. Omicron strains were comprised of 8 BA.1.1 (21 K), 27 BA.2 (21 L), and 1 BA.2.12.1 (22C). Phylogenetic analysis of the identified isolates and representative sequences of SARS-CoV-2 strains revealed clusters that presented the WHO VOCs. Specific or unique mutations for each VOC waxed and waned according to the variant waves. Our findings allowed recognition of the overall trends of SARS-CoV-2 isolates, which implicated replication advantage, immune evasion, and disease management.

## 1. Introduction

Coronavirus disease 2019 (COVID-19), which is caused by the severe acute respiratory syndrome coronavirus 2 (SARS-CoV-2), has spread worldwide. Accordingly, the pandemic has continued for over 2 years with the emergence of variants of concern (VOCs) [1,2]. By 1 April 2022, 6,142,735 cumulative deaths had been reported worldwide [3], which led to an enormous impact on social and healthcare systems. In South Korea, 14,001,406 cumulative confirmed cases and 17,453 deaths had been reported by August 2022, with outbreaks of variants [4].

Several evolving SARS-CoV-2 variants have rapidly emerged within populations, with different transmissibility and clinical implications compared with those of wild-type isolates. These variants have been assigned Greek alphabets by the World Health Organization (WHO) [5]. The Alpha variant was first reported in the United Kingdom in late 2020 and became the globally dominant variant until the emergence of Delta strains. The Alpha variant was approximately 50–75% more transmissible than previous strains [6,7]. The Delta variant (B.1.617.2 Pango Lineage) was first reported in India in December 2020 and became the most predominant variant worldwide, until the emergence of the Omicron variant [8]. Compared with previous VOCs, the Delta variant was more transmissible and associated with a higher risk of severe disease and hospitalization [9,10]. The Omicron variant, which was first reported in Botswana and shortly thereafter in South Africa in November 2021 [11], is also known as the B.1.1.529 lineage according to the Pango lineage, which is based on sequence relatedness and epidemiological events [12]. The variant harbors over 30 mutations in the spike protein, which contribute toward enhanced transmissibility and reduced susceptibility to neutralizing antibodies [13]. Therefore, the Omicron variant has caused the majority of SARS-CoV-2 infections in multiple countries, posing a severe threat to global healthcare systems [14,15]. Moreover, several sublineages of Omicron have been reported, including BA.1 and BA.2 [16,17,18], which became dominant variants with significantly reduced susceptibility to neutralization.

Few studies have tracked the variants causing the massive surges in South Korea. Therefore, we aimed to explore the evolution of SARS-CoV-2 strains by sequencing the spike protein over 29 months, which accounted for the majority of the COVID-19 pandemic period. Following genomic analysis, we classified the isolates according to the time frame, established the phylogenetic relationships, and identified the harboring mutations. Our findings could help elucidate the circulating variants and their potential influence on the control of COVID-19 surges.

## 2. Materials and Methods

### 2.1. Study Population and Sample Collection

A total of 109 nasopharyngeal swabs from patients with confirmed COVID-19 infection were randomly collected at Hallym University Dongtan Sacred Heart Hospital. The study population comprised 56 men and 53 women. The median age of the patients was 35.0 years (range: 0–90 years). Swab samples were collected from March 2020 to July 2022, which covered almost the entire period of the COVID-19 pandemic.

Data regarding the incidence of SARS-CoV-2 infection and related deaths from January 2020 to July 2022 were collected from the Korean Center for Disease Control (https://ncov.kdca.go.kr/, accessed on 1 September 2022). In 2021, the number of confirmed cases and COVID-19-related deaths was 9.4 and 5.2 times, respectively, higher than those in 2020. Moreover, the incidence and COVID-19-related death rates from January to July 2022 were 33.6 and 4.2 times, respectively, higher than those in 2021. This could be attributed to the loosening of preventive measures and the emergence of VOCs from SARS-CoV-2.

### 2.2. Sample Processing and Sequencing

The specimens were transferred to Universal Transport Medium (Copan Diagnostics, Inc., Brescia, Italy) and tested using RT-PCR assays. The QIAamp Viral RNA Mini Kit (Qiagen, Hilden, Germany) and QIAcube platform (Qiagen, Hilden, Germany) were used to isolate nucleic acids. STANDARD M SARS-CoV-2 (SD Biosensor, Suwon, Republic of Korea) targeting the envelope (E) and open reading frame 1ab (ORF1ab) genes, as well as the Allplex 2019-nCoV assay (Seegene, Seoul, Republic of Korea) targeting the E, ORF1ab, and nucleocapsid (N) genes, were used to detect SARS-CoV-2. Amplification and detection were performed using a Bio-Rad CFX96 thermocycler (Bio-Rad Laboratories, Hercules, CA, USA). All procedures were performed following the manufacturer’s instructions. A positive result was indicated by Ct values for all included genes within the predefined cutoff [19]. The median Ct values for the E and ORF1ab genes were 16.6 (10.3–26.1) and 17.2 (8.9–25.8), respectively.

The RNA extracts were stored at −70 °C until use. PCR and sequencing of the genes for spike proteins, including surrounding proteins such as nonstructural protein 16 (nsp16) and open reading frame 3a (ORF3a) proteins, were conducted using specific primers for eight fragments (Appendix A). Two additional primer sets were used for the first and third fragments in case of amplification failure of the first-round PCR [20]. After visualization of the PCR products using agarose gel electrophoresis, sequencing was performed using BigDye Terminator version 3.1 (Applied Biosystems, Foster City, CA, USA) and an ABI PRISM 3730XL Analyzer (Applied Biosystems, Foster City, CA, USA).

### 2.3. Sequence and Statistical Analysis

The Analyse-it method evaluation edition software version 2.26 (Analyse-it Software Ltd., Leeds, UK) was used for descriptive statistics. Continuous variables were presented using the median and range, while categorical variables were presented as the number with percentage. We compared the obtained sequences and reported Wuhan wild-type SARS-CoV-2 sequences using the Basic Local Alignment Search Tool (https://blast.ncbi.nlm.nih.gov/Blast.cgi, accessed on 1 September 2022) in the National Center for Biotechnology Information GenBank database and Stanford University Coronavirus antiviral and resistance database (https://covdb.stanford.edu/sierra/sars2/by-sequences, accessed on 1 February 2023) in order to identify VOCs of SARS-CoV-2 isolates based on genetic similarity. The gene sequences were analyzed according to the naming system, including the Pango lineage (https://pangolin.cog-uk.io/, accessed on 1 February 2023), Nextstrain clade (https://clades.nextstrain.org, accessed on 1 February 2023), and WHO label (https://covariants.org/, accessed on 1 February 2023). In addition, major mutations related to currently monitored variants (https://www.bv-brc.org/view/VariantLineage/, accessed on 1 February 2023) were analyzed.

Phylogenetic analyses were conducted to evaluate the genetic relationships among the SARS-CoV-2 sequences. The MEGA program version 11 was used to construct a maximum likelihood phylogenetic tree with 1000 bootstrap replications based on sequence alignment.

## 3. Results

### 3.1. Incidence and Mortality during the COVID-19 Pandemic

The first surge of COVID-19 cases in South Korea was reported on January 20, 2020 [21], with several waves of outbreaks being subsequently reported [22]. In 2020, 60,726 confirmed cases and 900 deaths were reported with the first, second, and third peaks (Figure 1 and Figure 2). Following a surge of cases that started in July 2021, 570,067 new cases and 4663 deaths were reported in 2021. This drastic increase could be attributed to the persistent emergence of new SARS-CoV-2 variants. In 2022, 19,144,315 confirmed cases and 19,483 deaths were reported between January and July 2022, which could be attributed to the infectivity of the Omicron variant.

### 3.2. Strain Classification

A total of 109 swab samples with Ct values < 30 on PCR analysis were collected. Among them, 3, 4, 32, 33, and 37 samples were collected from January to June 2020, July to December 2020, January to June 2021, July to December 2021, and January to July 2022, respectively. After gene sequencing and analysis, 34 wild-type strains and 75 VOCs, including 4 Alpha variants, 33 Delta variants, 2 Epsilon variants, and 36 Omicron variants were identified (Table 1). The Omicron strains detected included 8 BA.1.1 (21 K), 27 BA.2 (21 L), and 1 BA.2.12.1 (22C).

In 2020, wild-type SARS-CoV-2 isolates were collected. Alpha, Delta, and Epsilon variants emerged between January and June 2021 (Figure 3). Delta VOCs were predominant between July and December 2021 (the fourth COVID-19 wave). During the fifth wave (from January to July 2022), 97.3% of the isolated strains were Omicron VOC (21.6% of BA.1 and 75.7% of BA.2 as ongoing cases in South Korea). In comparison with a frequency plot from NextStrain (https://nextstrain.org/ncov/gisaid/global/all-time, accessed on 1 February 2023), Beta (20H), Gamma (20J), and Omicron (22A and 22B) was not found in South Korea. The starting points of Alpha (20I), Delta (21A), and Omicron (21K, 21L, and 22C) in South Korea were late by ≈ 2–3 months compared with the global averages.

### 3.3. Phylogenetic Relationships

The sequenced SARS-CoV-2 isolates were aligned and their phylogenetic relationships were evaluated. The resulting phylogenetic tree comprised of clusters presenting the WHO VOCs (Figure 4), with a cluster of wild-type isolates at the 9 o’clock position. A cluster of Alpha VOC for isolates in this study was observed at the 11 o’clock position. Near the Alpha cluster, the Epsilon cluster consisted of two isolates in this study and one strain of B.1.427, SARS-CoV-2 isolate. Delta strains collected after January 2021 were observed at the 2 o’clock position. The Omicron (BA.1 and BA.2) VOC cluster at the 6 o’clock position was distant from the other clusters. Representative sequences used for Wild, Alpha, Epsilon, Delta, and Omicron (BA.1, BA.2, and BA.2.12.1) types are presented in Appendix A.

### 3.4. Major Mutations Related to Variants of Concern

Major mutations in the spike proteins of VOCs are summarized in Appendix A. Three designated time frames (January to June 2021, July to December 2021, and January to July 2022) were adopted based on the time course distribution of VOCs. The most prevalent mutation was D614G, which was observed in 100.0% of Alpha, Delta, and Epsilon VOCs strains detected after January 2021. Among the representative spike protein mutations for Alpha, E484K, which has an established protein structure, accounted for 31.3% (10 out of 32) of mutations from January to June 2021. Other characteristic mutations of Alpha isolates included 501Y, P681H, H69del, and V70del. Most Delta mutations, including T19R, G142D, E156del, F157del, R158G, A222V, T478K, P681R, and D950N, showed an increase from July to December 2021 (from 6.3 [2 out of 32] to 90.9% [30 out of 33]) and a decrease from January to July 2022 (2.7% [1 out of 37]). K417N, which is a substitution mutation, was detected as the Delta variant evolved. S13I and W152C spike protein mutations specific for Epsilon were only observed from January to June 2021 (6.3% [2 out of 32]). Most mutations in Omicron variants appeared from January to July 2022; further, the differences between BA.1 and BA.2 are presented in Appendix A. The 13 mutations in BA.1.1 comprised ≈ 20% (7–14 out of 37) of the included strains. Meanwhile, mutations in BA.2 were found in ≈ 70% (26 to 28 out of 37) of the isolates. The unique mutations for BA.2.12.1 (22C), including L452Q and S704L, appeared in 2.7% (1 out of 37) of the strains from two isolates found between January and July 2022. Furthermore, the average numbers of mutations were 13.5 for Alpha, 8 for Epsilon, 12.3 for Delta, and 31.8 for Omicron VOCs, indicating the diversity of mutations of Omicron strains.

### 3.5. Uncommon Mutations According to the WHO Classification

We observed a novel E583D mutation in one isolate of the Alpha variant. Additionally, we observe three uncommon mutations (T29A, T299I, and Q613H) from two isolates and one unreported mutation (I834V) from one strain. For the Epsilon variant, one unreported mutation (A623S) was identified from one isolate. An uncommon mutation (F486V) was found in an Omicron BA.2 isolate. Finally, one unreported mutation (C1243F) was detected in BA.1, while three unreported mutations (L110S, N925K, and E1182Q) were detected in BA.2 isolates.

## 4. Discussion

This study assessed the distribution of the SARS-CoV-2 strains in South Korea from March 2020 to July 2022. During the COVID-19 pandemic, five surge peaks were observed in South Korea. Among them, the fifth wave, which was caused by the Omicron VOC, had the highest peak of incidence. Wild-type strains were only found in samples collected in 2020. From January 2021, VOCs such as Alpha (B.1.1.7, 20I), Delta (B.1.617.2, 21A), Epsilon (B.1.427, 21C), and Omicron were identified. Omicron strains, which were predominant after January 2022, comprised of BA.1.1 (21 K), BA.2 (21 L), and BA.2.12.1 (22C). Phylogenetic analysis of representative strains revealed clusters of wild-type, Alpha, Delta, Epsilon, Omicron (BA.1), and Omicron (BA.2) strains. Among the major mutations in the spike protein of VOCs, D614G was the most prevalent. Specific or unique mutations of each VOCs waxed and waned according to the wave of SARS-CoV-2 variants.

The emergence and genomic diversity of SARS-CoV-2 in different countries have been reported [23]. The first case of COVID-19 in China was reported on 19 January 2020 [24]. Following a surge in February 2020, large-scale surveillance and intervention measures were implemented [25]. After intensive testing and contact tracing, the number of COVID-19 cases decreased, which demonstrated the importance of surveillance. Alpha and Beta VOCs were identified by the Center for Disease Control (CDC) in January 2021 [23]. The first case of international importation in the USA was reported on 20 January 2020 [26]. Multiple importations of SARS-CoV-2 from China, Europe, and other parts of the USA led to subsequent large outbreaks in Washington State, New York City, and Northern California [27,28,29]. Genomic surveillance allowed clear illustration of the transmission of SARS-CoV-2, and highlighted the importance of intensive testing and contact tracing in controlling the spread of SARS-CoV-2. In South Korea, the first case was reported on 20 January 2020. Several detection kits, such as Allplex 2019-nCoV and STANDARD M nCoV, have been developed in South Korea [30]. However, the status of SARS-CoV-2 strains during the COVID-19 pandemic based on sequencing analysis remains unclear. In this study, we sequenced and analyzed the samples collected during the COVID-19 pandemic. In 2020, 2021, and 2022 (from January to July), 7, 65, and 37 cases were sequenced, respectively. Accordingly, the number of analyzed cases varied across the years.

Genomic data for VOCs as well as wild-type SARS-CoV-2 are deposited in several databases, including the Global Initiative on Sharing All Influenza Data, the National Center for Biotechnology Information GenBank, and the Sequence Read Archive. Moreover, the Pango lineage, Nextstrain, and WHO nomenclature systems have been applied for SARS-CoV-2 genomes [12]. The obtained sequences can be assigned to the most likely lineages or clades using these systems.

The Alpha variant, which we detected in 2021, was first identified in the United Kingdom in late 2020. It became the globally dominant variant until the emergence of the Delta variant [23]. This variant was assigned to B.1.1.7 in the Pango lineage and 20I in the NextStrain clade. It is defined by 17 amino-acid-altering mutations, including eight mutations in the spike protein. The E484K mutation in the receptor-binding domain is a major mutation that has been well-characterized [31]. Additionally, N501Y, P681H, H69del, and V70del mutations had potential biological relevance. It showed an increase of approximately 50–70% in infectivity and a substantial risk of death [6,32]. Despite the moderate attenuation of vaccination effects, B.1.1.7 remained sensitive to neutralization and was unlikely to be a major concern for vaccination resistance or an increased risk of reinfection [33,34].

Delta variants increased from January 2021 to July 2022 and became predominant from July to December 2021 (90.9%) in South Korea. This variant was first detected in India in December 2020 and was the most prevalent variant in European countries by April 2021, prior to its detection in South Korea [31]. This variant was assigned to B.1.617.2 according to the Pango lineage, and comprised 21A, 20I, and 21J in the NextStrain clade. Delta strains harbor several important amino acid mutations in the spike protein, including E156del, F157del, R158G, and T478K mutations [23]. K417N is recognized for its increased transmissibility, risk of hospitalization, and burden on healthcare services [35,36]. There was a slight decrease in vaccine effectiveness among symptomatic patients with the Delta variant; however, the effectiveness remained high against severe disease and hospital admissions [10,37]. Among the uncommon mutations, T29A and T299I from two Delta isolates have been identified in Morocco [38]. T29A and T299I mutations result in increased molecule flexibility. Specifically, T29A has a destabilizing effect and alters the structure of the SARS-CoV-2 spike N-terminal domain. Q613H, another uncommon mutation, has been detected in Sweden and Pakistan [39,40].

The Epsilon variant was first identified in May 2020. It caused >50% of cases detected in California from September 2020 to January 2021 [41]. In South Korea, cases caused by the Epsilon variant were observed in February and April 2021. The CDC and WHO originally designated it as a VOC in early March 2021. After further monitoring, it was designated as a variant of interest on 29 June 2021 [31]. This variant comprised B.1.427 and B.1.429 in the Pango lineage and 21C in the NextStrain clade. Compared with the wild-type strain, this variant had an 18–24% increase in transmissibility [41,42]. The specific mutations in this variant were S13I and W152C in the N-terminal domain. Compared with the wild-type strain, the Epsilon variant showed a three-fold decrease in neutralizing titers after vaccinations due to remodeling of the peptide cleavage site and formation of a disulfide bond derived from S13I and W152C mutations [43].

The Omicron variant originated in South Africa and caused a dramatic surge of cases in many other countries. Our findings indicated that this variant emerged after January 2020 in South Korea. Omicron sublineages with greater replication advantages have replaced the previously predominant sublineage. The first identified Omicron variant was BA.1, followed by BA.2. 21 K and 21 L, which were assigned as BA.1 and BA.2 in the NextStrain clade, respectively. The average number of BA.1 and BA.2 Omicron variants was 39 and 31, respectively [44], which is similar to the value in our study (31.8). The number of mutations was higher than that of previous VOCs. Since these mutations, Omicron variants have a higher positive electrostatic surface potential, leading to a higher affinity of the receptor-binding domain of human angiotensin-converting enzyme 2. Moreover, Omicron VOCs have been associated with a higher reinfection rate than Delta VOCs [45,46]. Among the Omicron sublineages, BA.2 has a higher transmission potential than BA.1. The difference in binding affinity may be attributed to the difference in mutations affecting the protein structure [44,47]. These replication advantages of the Omicron sublineages may be associated with immune escape. In some studies, sera from participants with prior infection or vaccination did not neutralize Omicron variants [17,48]. Furthermore, monoclonal antibody therapies, including casirivimab/imdevimab and sotrovimab were inactive against BA.2, which highlights the importance of monitoring VOCs [49,50]. The uncommon mutation F486V, from two Omicron BA.2 isolates, has been detected in the BA.4/5 sub-variants. The alteration in F486V may result in increased antibody evasion; accordingly, it is associated with reduced neutralization [51,52]. Further research on the unreported mutations, including E583D from Alpha; I834V from Delta; A623S from Epsilon; C1243F from Omicron BA.1; and L110S, N925K, and E1182Q from Omicron BA.2 are warranted to reveal their roles regarding the survival of SARS-CoV2 isolates.

This study has several limitations. First, this study had a relatively small sample size, especially in 2020. Further long-term large-scale studies are warranted to allow accurate surveillance of variants. Additionally, detailed functional interpretations, including effects on diagnostic tests and neutralizing antibodies, are warranted.

## 5. Conclusions

Several waves of incidence and deaths after the first case in January 2020 were caused by wild-type and variants of SARS-CoV-2, based on its vulnerability to mutations as a single-stranded RNA virus. Detecting and monitoring the evolution of SARS-CoV-2 variants is important, since the replication advantage, immune evasion, and disease severity could vary among these variants. This study described the temporal changes in SARS-CoV-2 isolates in South Korea from March 2020 to July 2022. After detection of the wild-type in 2020, Alpha (B.1.1.7, 20I), Delta (B.1.617.2, 21A), Epsilon (B.1.427, 21C), and Omicron VOCs were detected. Since January 2022, Omicron subvariants such as BA.1.1 (21 K), BA.2 (21 L), and BA.2.12.1 (22C) have been reported. The phylogenetic tree showed distinct clusters of wild-type, Alpha, Delta, Epsilon, Omicron (BA.1), and Omicron (BA.2) variants. Major mutations in VOCs waxed and waned according to the wave of SARS-CoV-2 variants, indicating the clinical significance of genomic surveillance.

## Figures and Tables

**Figure 1 viruses-15-00873-f001:**
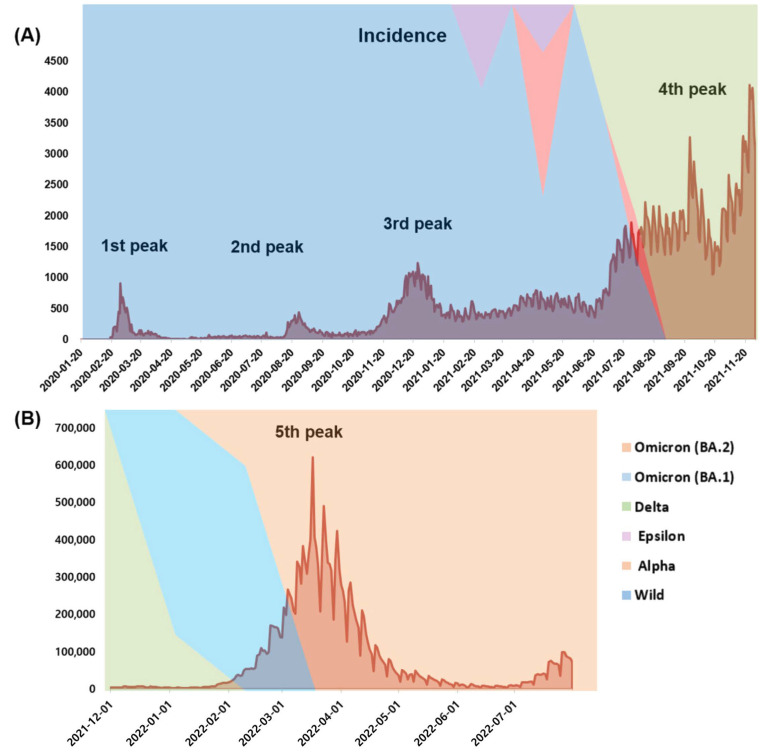
Distribution of newly confirmed cases of SARS-CoV-2 with five surge peaks in South Korea (**A**) from January 2020 to November 2021 and (**B**) from December 2021 to July 2022. Data from the Korean Center for Disease Control (https://ncov.kdca.go.kr/, accessed on 2 January 2023).

**Figure 2 viruses-15-00873-f002:**
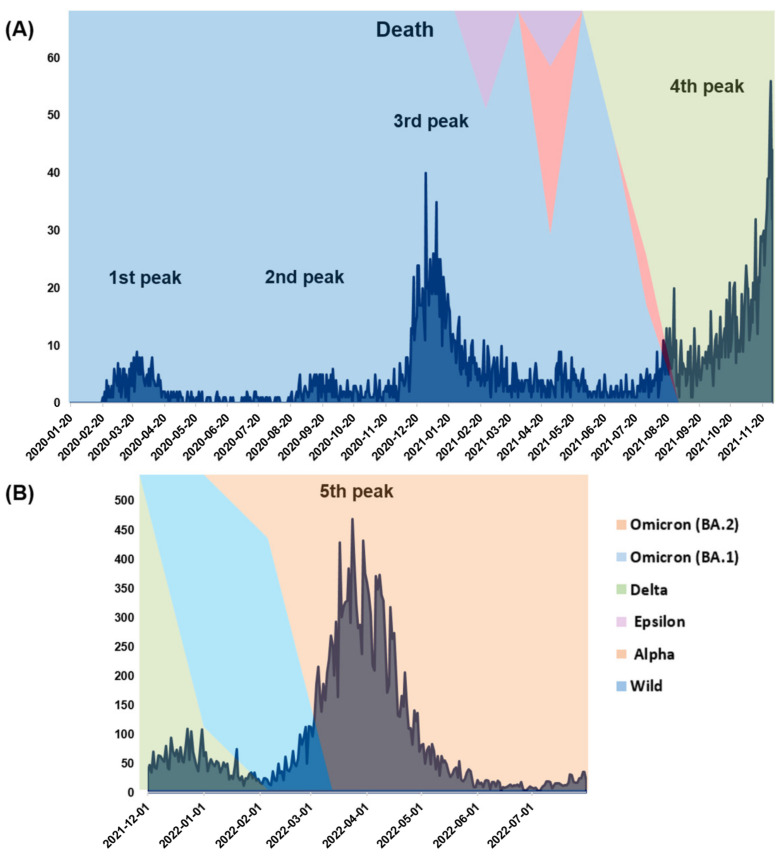
Distribution of deaths related to SARS-CoV-2 with peaks corresponding to the incidence peaks in South Korea (**A**) from January 2020 to November 2021 and (**B**) from December 2021 to July 2022. Data from the Korean Center for Disease Control (https://ncov.kdca.go.kr/, accessed on 2 January 2023).

**Figure 3 viruses-15-00873-f003:**
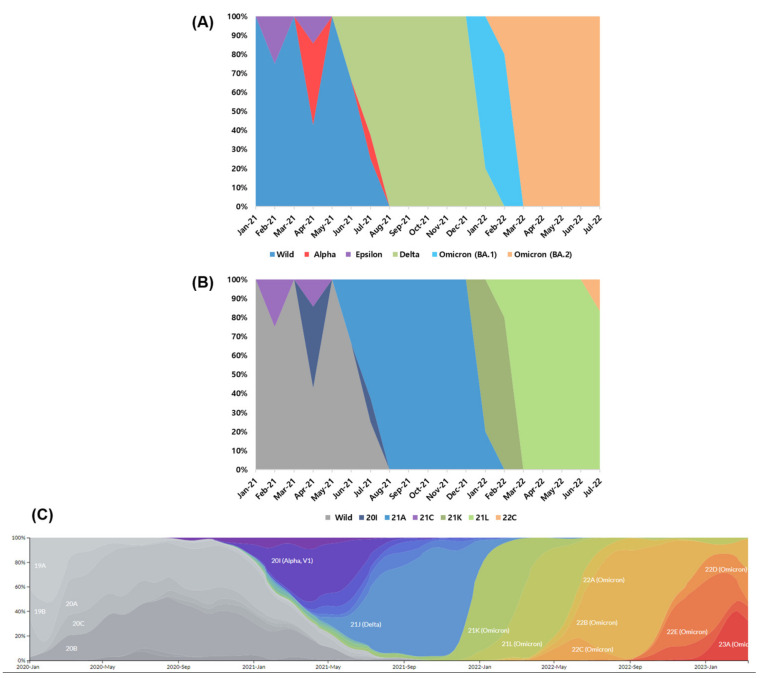
Time course distribution of sequenced SARS-CoV-2 strains in South Korea between January 2021 and July 2022. (**A**) The World Health Organization (WHO) and (**B**) NextStrain classification systems were applied. (**C**) A plot including approximately 4000 genomes collected from January 2020 to March 2023 globally (https://nextstrain.org/ncov/gisaid/global/all-time, accessed on 1 February 2023).

**Figure 4 viruses-15-00873-f004:**
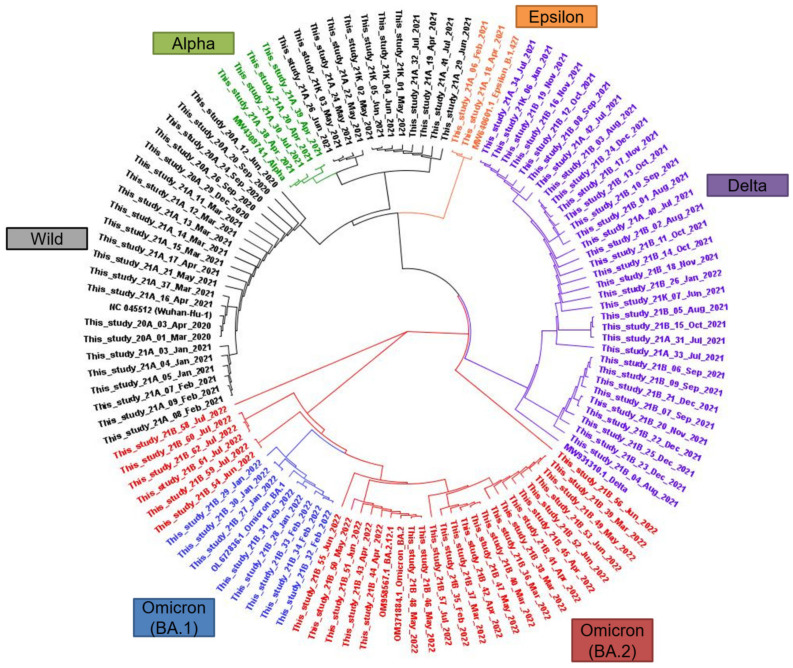
Circular phylogenetic tree representing the relationship of sequenced SARS-CoV-2 strains in South Korea between January 2020 and July 2022.

**Table 1 viruses-15-00873-t001:** Distribution of SARS-CoV-2 strains for 29 months in South Korea.

Time	Wild	Alpha	Delta	Epsilon	Omicron (BA.1)	Omicron (BA.2)	Omicron (BA.2)
		21I	21A	21C	21K	21L	22C
		B.1.1.7	B.1.617.2	B.1.427/B.1.429	BA.1.1	BA.2	BA.2.12.1
2020.01–2020.12	100% (7/7)	0% (0/7)	0% (0/7)	0% (0/7)	0% (0/7)	0% (0/7)	0% (0/7)
2021.01–2021.06	78.1% (25/32)	9.4% (3/32)	6.3% (2/32)	6.3% (2/32)	0% (0/32)	0% (0/32)	0% (0/32)
2021.07–2021.12	6.1% (2/33)	3.0% (1/33)	90.9% (30/33)	0% (0/33)	0% (0/33)	0% (0/33)	0% (0/33)
2022.01–2022.07	0% (0/37)	0% (0/37)	2.7% (1/37)	0% (0/37)	21.6% (8/37)	73.0% (27/37)	2.7% (1/37)

## Data Availability

The data used and presented in this study are deposited at https://dataverse.harvard.edu/, accessed on 1 February 2023 (doi.org/10.7910/DVN/MA1O9C, accessed on 20 March 2023) and the GenBank under accession number from OQ380945 to OQ381045.

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
