# Peer review of "Tracking the Genomic Evolution of SARS-CoV-2 for 29 Months in South Korea"

_viruses, 2023, doi:10.3390/v15040873_

Round 1
Reviewer 1 Report
In its basic form, this manuscript presents a technically sound analysis of a small set of SARS-CoV-2 cases in South Korea. It is generally well written and includes sufficient background information to contextualize the results. However, the results have little scientific novelty, reflecting expected global trends and timelines.
The best way to improve this manuscript given the data available is to perform more direct comparisons to global trends in VOC prevalence. Finding that a specific VOC spread faster or slower within South Korea as compared to other nations is of public health interest. Global genomic surveillance data is abundant and can be obtained from many sources.
Minor comments:
Figures 1 and 2 do not present any of your novel genomic data in any form. Coloring the background with hues reflecting the predominant strain in South Korea as estimated from your data would inform readers as to the correlation of your genomic surveillance with these death reports and reinforce the importance of your work.
You may consider replacing Figure 3 with a frequency plot ala a Nextstrain narrative (https://nextstrain.org/ncov/gisaid/global/6m), particularly if you rework this section with global VOC prevalence as context.
The 3.3 results section is clogged with genbank accession information that makes it difficult to read. The authors should move sample accession information to a supplementary table instead of writing it out in the text. There are minor English language issues in this section as well. This section could generally be rewritten to be less verbose or even removed, as it mostly repeats what is visually obvious in Figure 4.
Figure 4, similar to Figure 1 and 2, would be made more visually pleasing if overlaid over piechart-like slices of color corresponding to the clades in question.
Table 2 would be better as supplemental material, as it takes up substantial page space while contributing relatively minimal information. Readers interested in frequency of specific VOC-related mutations can inspect the supplement.
Lines 249-254 should be in the Methods (2.3), not the Discussion.
Lines 255-306 feel organizationally misplaced, and should be included as a Section 3.5 in the Results, with some additional sentences summarizing any unique patterns for that VOC in South Korea found in your dataset. This section could also be generally reduced, as you do not need to include full context for why each VOC is important for your conclusions. The discussion could include some of this VOC-specific context, though, if you draw a particular connection between VOC behavior in South Korea and established VOC characteristics.
Author Response
We would like to thank Reviewer 1 for your time and efforts in reviewing our manuscript and for providing comments, which have considerably helped us improve our manuscript. We have made revisions based on your comments and have provided our point-by-point responses below. We hope that our responses and revisions appropriately address your comments.
We attached the file (Response to reviewers' comments)

Reviewer 2 Report
Introduction section. Line 50-54. The cited references are not relevant. Either the authors should modify the sentences while mentioning the countries like India and Botswana or they should cite the relevant references. For example for the emergence of Delta VOC in India, the author should have cited <Genes 2021, 12(11), 1803; https://doi.org/10.3390/genes12111803>.
Similarly, due credit should be given to scientists from Botswana and Africa who identified the new VOC.
Methods and Results: My major concern is about the very few numbers of strains collected/processed over the long duration of more than two years. Specifically, line numbers 141-144. Where in only 3 and 4 strains were isolated in 6 monthly periods of Jan -June 2020 and July -Dec 2020 period. Therefore, these small numbers are insufficient to arrive on any conclusion. Preferably, these 7 samples should have not been included in this study.
Author Response
We would like to thank Reviewer 2 for your time and efforts in reviewing our manuscript and for providing comments, which have considerably helped us improve our manuscript. We have made revisions based on your comments and have provided our point-by-point responses below. We hope that our responses and revisions appropriately address your comments.
We attached the file (Response to reviewers' comments)

Reviewer 3 Report
1.Nicely written article wherein authors have tracked the genomic evolution of SARS-C0v-2.
2.Sample size is less and when samples are stratified into different period the size very less particularly Jan to June and July to Dec 2020 .
3.The results of mutations are expressed in percentages numbers of needs to be mention(out of how many)
Author Response
We would like to thank Reviewer 3 for your time and efforts in reviewing our manuscript and for providing comments, which have considerably helped us improve our manuscript. We have made revisions based on your comments and have provided our point-by-point responses below. We hope that our responses and revisions appropriately address your comments.
We attached the file (Response to reviewers's comments)

Reviewer 4 Report
In the current manuscript, the authors have sequenced the SARS-CoV-2 virus in patients, over a two year period, and analyzed the occurence and proportions of different variants.
The study is a valuable resource for documenting the dynamics of the pandemic-causing virus, and correlating the effects of variants with the number of cases and deaths.
The study is observational, straightforward in design, and well-executed. The methods and analysis are mostly well described. However, the authors need to address a few issues as listed below.
1. Since the study involves human subjects, the authors should ensure that all relevant ethical guidelines and approvals are followed and reported as per: https://www.mdpi.com/journal/viruses/instructions#ethics .
2. The authors have not sequenced the entire genome, but three gene regions from the SARS-Cov-2 virus, namely spike, NSP16 and ORF3a. Therefore statements such as in line 66 "SARS-CoV-2 strains at the genomic level", or line 113 "The genome sequences were analyzed", or line 162 "The genomes of the sequenced SARS-CoV-2 isolates" are not accurate. Please change to "gene sequence analysis" or equivalent.
3. In Line 128 : "Despite vaccination, mask wearing, and social distancing, variants of SARS-CoV-2 persistently contribute to the drastic increase in COVID-19 cases", the first half of the sentence is not necessary as the study is not evaluating the effects of vaccines or masks or social distancing. Without getting into those issues, the authors can simply state that "The variants of SARS-CoV-2 persistently contribute to the drastic increase in COVID-19 cases" or similar.
4. The source of data used to plot Figure 1 and Figure 2 should be clearly mentioned in the text as well as in the figure legend.
5. Figure 1 is confusing and the legend carries very little explanation on how to read the figure. It appears that due to the dramatically high surge in cases in 2022-03-20 window around 5th peak, the previous 4 peaks are not visible on the linear graph. Therefore the authors have overlayed separate panels for the first 4 peaks on the full time-series. It would be better to make a separate panel for the first 4 peaks, with a shorter y-axis up to 5,000 cases. More details should be added to the legend to guide the reader. Another option to represent such widely varying data in a single graph is to log-transform the data, i.e. plot the log10(Number of cases) on y-axis.
Also, the blue background of the figure is unnecessary and not a good data visualization practice. It could be left as plain white.
6. For Figure 2, see above comments for Figure 1 and modify accordingly.
7. Table 1 would be better presented by switching the rows and columns. The independent variable (time windows) should preferably be along the rows, and the observations (% of variants) should be in columns. Another column, showing the total number of samples analyzed in each time-window should also be added. Also, when the variant is at 100%, no decimal points should be added - please change "100.0%" to "100%".
8. Figure 4: The legend is incorrect as it displays a phylogenetic tree, not a time series (which would have time values along x axis e.g. as in Figure 3). Also, some variants show up at multiple time-windows, therefore the phylogenetic clades are not clustered by date. The difference between the Omicron BA.1 versus BA.2 clades is not easy to see. The authors can use colored labels, or background highlights to distinguish the clades for different variants. This can be done using any tree editing software such as FigTree (http://tree.bio.ed.ac.uk/software/figtree/).
9. It might be simpler for the reader if the full table is split into separate tables corresponding to each variant. A column/row showing the total, absolute number of samples analyzed in each category should also be added. If the authors can flip the rows and columns as suggested for Table 1 above, it would also improve the readability of the table. Finally, since the percentages of all alleles of. Variant (e.g. S13I, W152C and D614G for Epsilon) will not add up to 100%, an additional row/column for "Other alleles" could be added to make this more clear.
10. As mentioned earlier, "100.0%" should be changed to "100%" in Table 2 as well.
11. The link to the Harvard dataverse item doi.org/10.7910/DVN/MA1O9C is currently not working. Please provide a functional link with publicly available data.
*-*-*
Author Response
We would like to thank Reviewer 4 for your time and efforts in reviewing our manuscript and for providing comments, which have considerably helped us improve our manuscript. We have made revisions based on your comments and have provided our point-by-point responses below. We hope that our responses and revisions appropriately address your comments.
We attached the file (Response to reviewers' comments)

Round 2
Reviewer 1 Report
The authors have generally sufficiently addressed my review.
Reviewer 4 Report
The authors have revised the manuscript substantially and addressed all the comments from the previous round of reviews. The manuscript can be accepted.